# Observational Prospective Study to Determine the Efficacy of ‘Non-Slip Socks’ vs. ‘Adequate Footwear’ Regarding the Number of Falls Observed among Admitted Patients

**DOI:** 10.3390/healthcare11192605

**Published:** 2023-09-22

**Authors:** Rosa M. Moreno Rodríguez, Beatriz Solas Gómez, Laura Gallego Marcuello, Maria del Carmen Diaz Martinez, Encarnacion Fernández del Palacio, Andrés Santiago-Sáez

**Affiliations:** 1Nursing Department, Hospital Clínico San Carlos, 28040 Madrid, Spain; beatriz.solas@salud.madrid.org (B.S.G.); lgallegom@salud.madrid.org (L.G.M.); efpalacio@salud.madrid.org (E.F.d.P.); 2Geriatric Service, Hospital Clínico San Carlos, 28040 Madrid, Spain; mdiazmartinez@salud.madrid.org; 3Head of Legal Medicine Department, Hospital Clínico San Carlos, 28040 Madrid, Spain; 4Instituto de Investigación Sanitaria (IdISSC), 28040 Madrid, Spain; 5Facultad de Medicina, Universidad Complutense de Madrid, 28040 Madrid, Spain

**Keywords:** healthcare, risk, hospital, prevention

## Abstract

Background: Fall prevention is an important indicator of the quality of patient care. Prevention includes the use of adequate footwear. Our objective is to determine the differences in the number of falls between patients with “adequate footwear” and “non slip socks”, and their associated consequences, to support their use in the prevention of falls among hospitalized patients. Methods: This is an observational prospective study on inpatient falls. Patient characteristics, fall circumstances, and injuries were collected through Clinical Report Forms, a review of fall reports, and medical records. Admitted patients over 18 years old were recruited from Geriatric and Internal Medicine Units over a brief period of 3 months. Results: A total of 158 hospitalized patients were recruited. In total, 77 patients (48.73%) were assigned to the non-slip socks group, and 81 (51.27%) were assigned to the adequate footwear group. There were 21 falls during the study period, all of which were experienced by the adequate footwear group (*p* < 0.0001). The mean age of the patients who fell was 83.14 (range 60–100) years old. The most frequent reasons for admission among the patients who fell were COVID-19 infection (19%) and oncological complications (19%). Overall, 61.9% of patients had a high risk of falling. Most falls (76.1%) occurred in patient rooms, and most of these occurred while wandering around. The most frequent reason for falls was slipping (14/21). For 16 of 21 patients, falls did not have immediate consequences, while 5 had contusions and 1 suffered a wound. Nobody needed to be admitted to the ER or suffered external hemorrhages or loss of consciousness. Conclusions: Non-slip socks represent an adequate alternative to well-fitting rubber-soled footwear. It seems that non-slip socks could prevent falls among hospitalized patients; nevertheless, further studies are necessary to clarify their role in preventing hospital falls and reducing injury rates.

## 1. Introduction

The World Health Organization (WHO) defines falls as “involuntary events that cause you to lose your balance and find the body on land or another firm surface that stops it”. According to the WHO, falls are a major global public health problem. An estimated 684,000 fatal falls occur annually, making this problem the second leading cause of death from unintentional injury worldwide after road traffic collisions [1]. Falls are adverse events that also occur in the hospital environment. Hospital falls remain a problem worldwide, despite sustained fall prevention efforts in public and private healthcare settings [2,3].

Worldwide fall rates, which are usually expressed per 1000 bed days, typically range from two to eight in acute hospitals, geriatric wards, and emergency rooms [4,5].

According to data from the National Health Service (NHS), one in three individuals over the age of 65 will suffer at least one fall a year, and this proportion increases to one in two individuals for those over the age of 80 [6].

In Spain, the incidence of falls in acute hospitals shows significant differences depending on the characteristics of each hospital [7,8,9,10,11,12,13,14,15]. The mean incidence of falls in acute hospitals is estimated to be 1.66% with respect to the number of patients admitted, which corresponds to a total of 29,886 falls [16]. The percentage of falls reaches 6.3% among patients over 65 years old [17]. The consequences of falls often have a major impact on the well-being of a patient and their family as well as on the costs and sustainability of the healthcare system. The approximate mean cost of these falls is EUR 14.3 million [16].

In 2020, the incidence of falls in the hospitals of the Madrid Health Service was 1.76%. In total, 75% of hospitals had an incidence of falls of less than 2.38%, with a minimum result of 0.20% and maximum of 4.54%. The rate of falls varies depending on the level of complexity of a hospital. The increase in the incidence of falls compared to 2019 is related to the critical situation caused by COVID-19 [18]. The incidence rate of falls among hospitalized patients in the Madrid autonomous community was calculated per 1000 days of stay during the study period and included adult and pediatric patients from different services, such as medical, surgical, mental health hospitalization, and resuscitation services in intensive care units (adults and pediatrics) and maternity, obstetrics, and delivery services. Taking into account the average days of stay in Spanish hospitals (8.75 days), the incidence of falls in Madrid translated to % is similar to that of other hospitals in Spain [19].

The consequences of falls for an individual and for the health system must be added to this high incidence. Falls considerably increase morbidity and mortality among the most vulnerable population, as they are the most common source of injuries among the elderly. While most falls inflict minor damage, approximately 30% of individuals will suffer injuries of different considerations, and between 4% and 6% of patients suffer serious injuries, including fractures, hemorrhages, or subdural hematomas that, in severe cases, threaten the life of an individual [20].

Falls among hospitalized older adults are also associated with a longer hospital stay and poor patient outcomes [3]. Falls have multifactorial etiologies that are associated with intrinsic risk factors (age, sex, level of consciousness, sensory or motor alteration, alteration in mobility or elimination, consumption of drugs, and previous falls, among others) and extrinsic risk factors (lighting, state of the ground, obstacles, footwear, aids for ambulation, etc.).

Fall prevention is an important indicator of the quality of patient care.

It is advisable to promote safety strategies aiming to reduce the incidence of falls for patients during their admission to a hospital. Prevention strategies include patient education [21,22,23], bed alarms [24], assistive devices, exercise, rehabilitation [25,26], medication reviews, fall policies and systems, and environmental adaptations [3].

Prevention includes the use of adequate footwear. Patients often have inadequate footwear at the time of their arrival to a hospital or during their stay [27]. Inadequate footwear is associated with falls and fall-related fractures [28].

Non-slip socks are sometimes provided to hospitalized patients to reduce their risk of falling [27,29,30]. Non-slip socks, also known as ‘traction socks’ or ‘grip socks’, have a rubber or synthetic tread that provides friction with the aim of increasing stability [31,32]. Safe footwear also helps to guard against infection when walking and enables people to move more safely [33,34].

The literature on non-slip socks as a footwear alternative has yielded mixed findings [31,35,36]. Some have claimed benefits, while others have reported that walking bare-footed offers similar slip resistance [32].

In order to clarify the role of non-slip socks, we planned an observational study with the aim of determining the differences in the number of falls between patients with “adequate footwear” and “non slip socks”, and the associated consequences, to support their use in the prevention of falls among hospitalized patients.

## 2. Materials and Methods

### 2.1. Study Population

An observational study was conducted in “Hospital Clinico Universitario San Carlos” (a third-level hospital) located in Madrid. Ethics approval was sought and gained from the Human Research Ethics Committee.

Admitted patients over 18 years old were recruited from Geriatric and Internal Medicine Units between March 2022 and June 2022.

Non-slip socks were assigned to patients with unsuitable footwear [Figure 1]. The appropriate footwear was determined according to the following criteria: the degree to which the type of footwear was flexible, lightweight, and provided heel support; had soles that were not smooth but instead embossed to provide a non-slip texture; had closed heels; were non-laced; and were Velcro-fastened.

### 2.2. Evaluation of Patients

A Clinical Report Form [CRF] was created, which included the following variables: age, gender, admission unit, reason for admission, data from the Spanish version of the Downton Fall Risk Index score [used in the admission process] such as known previous falls, medications [tranquillizers/sedatives, diuretics, antihypertensives (other than diuretics), antiparkinsonian drugs, antidepressants, and others], sensory deficits such as visual and hearing disturbances, limb impairment, mental status [orientated or confused], and gait [safe without walking aids, safe with walking aids, unsafe with or without walking aids, or unable]. A score over 3 was considered to correspond to a high risk of a fall. The CRF also included data associated with a given fall, such as the site, reason [sliding, handrail failure, bed brake failure, and stumbling over furniture], associated symptoms [seasickness, syncope, loss of strength, hypoglycemia, hyperglycemia, hypotension, oxygen saturation, delirium, acute confusional syndrome, and others], other circumstances as urinary or fecal incontinence, tubes, drain-related mobility problems, changes in treatment in the last 48 h, background of falls in the last 6 months, insufficient lighting, non-slip floors, environmental obstacles or disorder, and available assistive devices, among others. Consequences of the fall were also registered, such as contusions, wounds, external bleeding, loss of consciousness, and ER admission. In addition, fall reports and medical records related to an episode were reviewed.

The questionnaires were collected by trained nurses.

Two groups were compared: patients with suitable footwear and patients with non-slip socks.

### 2.3. Statistical Analysis

Based on the prevalence of falls reported in Spain and the statistics in our hospital, in the same services during the same period in previous years, the prevalence of falls was estimated to be 1.1%. Assuming an alpha value of 0.05 and a margin of error of 5%, the required sample size was estimated to be 150 subjects.

Patient demographic, clinical, and documented comorbidity characteristics were compared between patients who received non-slip socks and those with suitable footwear. Descriptive analyses were assessed. Student’s *t*-test, a Mann–Whitney test used to compare the means (of quantitative variables), and a Chi-square test of proportions (of categorical variables) were conducted. A *p*-value of 0.05 was used as a marker of statistical significance. Analyses were performed with statistical software XLSTAT version 2022.4.1 Addinsoft (2023) (XLSTAT statistical and data analysis solution. New York, NY, USA, https://www.xlstat.com/es [accessed on 1 September 2023].

## 3. Results

A total of 158 hospitalized patients were recruited. In total, 77 patients [48.73%] were assigned to the non-slip socks group, and 81 [51.27%] were assigned to the adequate footwear group. There were no statistical differences between the groups in terms of gender; previous treatments; antihypertensives, psychotropics, anticoagulants, and others; visual and hearing disturbances; limb impairment; and previous risk of falls.

Statistical differences were observed for age (76.39 ± 17.55 for the non-slip socks patients vs. 84.45 ± 10.75 in the adequate footwear group [*p* < 0.001]) and in terms of mental status (12 patients had mental confusion in the non-slip socks group vs. 4 among the adequate footwear patients [*p* < 0.027]).

The most frequent reason for admission among patients in the non-slip socks group was respiratory infections [24.26%] followed by heart failure [15.58%] and urinary infection [6.5%]; for the patients with adequate footwear, the main reason was heart failure [18.51%] followed by respiratory [13.58%] and urinary infections [9.8%]. The basal characteristics of the patients are summarized in Table 1.

There were 21 falls among 158 patients during the study period. All of them occurred in the adequate footwear group [*p* < 0.0001]; 137 patients did not suffer falls, i.e., all 77 patients in the non-slip socks group and 60 in the adequate footwear group.

The following analyses were carried out based on the data regarding the time of each fall. Of the 21 falls, 14 were suffered by males and 7 were suffered by females. The mean age of the hospitalized patients who fell during admission was 83.14 [range 60 to 100] years; the most frequent reasons for admission among patients with falls were COVID-19 infection [19%] and oncological complications [19%], followed by heart failure [14.28%] and respiratory [9.5%] and urinary infections [9.5%]. Of 21 patients, 13 [61.9%] had a high risk of falling, 7 [33.3%] had a medium risk, and 1 did not have a risk of falling. The patients’ risk factors for falls are summarized in Table 2.

Most falls [16 falls, 76.1%] occurred in the patients’ rooms, and most of these occurred while wandering around [7/21 falls, 33.33%], while 6 occurred [28,5%] while trying to sit down and 6 [28.5%] took place at the bedside.

The most frequent reason for falls was slipping [14/21], followed by loss of strength [6/21 falls] and acute confusional syndrome [3/21]. Other events included, stumbling over furniture, dizziness, handrail failure, and others. No falls were related to hypoglycemia, hyperglycemia, hypotension, or a low saturation of oxygen (Figure 2).

Of 21 patients, 8/21 had urinary incontinence, 6/21 had fecal incontinency, 7/21 had mobility disorders, 1/21 had catheters, 2/21 had inserted drains, and 5/21 had changed their medication in the last 48 h.

Regarding environmental risk factors, 85.7% of the patients had enough lighting, 57.1% had a non-slip floor, 90.5% did not encounter environmental obstacles, 95.2% had a bed with a suitable height, 90.5% had assistive devices at their disposal, 61.9% had an adapted bathroom with handholds, 52.4% did not wear closed-heel shoes, 61.9% had a bed with handrails, 95.2% had an accessible ringtone, 9.5% had a chest fastening strip, and 4.7% had an abdominal fastening strip.

For 16 of the 21 patients, falls did not have immediate consequences, while 5 had contusions and 1 suffered a wound. Nobody needed ER admission, suffered external hemorrhages, or experienced a loss of consciousness.

A logistic regression was performed to model the effect of some of the variables on falls. We considered two explanatory variables for falls, one qualitative—type of footwear—and one quantitative, corresponding to age. When the Chi^2^ associated with the Log ratio is lower than 0.0001, we can conclude that the variables provide a significant amount of information. In the Type II analysis, we can see that the variable that most influences falls is the type of footwear (*p* = 0.000). The use of “nonslip socks” had a significant negative impact (−18.102) on falls, whilst age did not have a significant effect. The classification table for the training sample shows that the observations were well classified at 86.71%. The area under the curve for the ROC curve was 0.814, showing a very discriminating model (Figure 3).

The effect size between falls and type of footwear showed a weak association between variables (Pearson correlation coefficient and Cramér’s V coefficient = 0.382).

## 4. Discussion

In our study, no falls were registered in the non-slip socks group, showing that non-slip socks had a significant negative impact on falls, which aligns with the findings of Pelliard et al.’s systematic review, in which it was concluded, based on seven studies, that non-slip socks had a preventive effect on reducing the recurrence of falls [*p* = 0.009] [37], while contrasting with the reviews conducted by Jazayeri [38] and Hartung and Lalonde [31], who did not find sufficient evidence to support the use of non-slip socks to prevent falls in hospitals. Certainly, to the best of our knowledge, there has only been one interventional study that assessed the effects of non-slip socks on fall rates [39]; this small trial of 39 patients showed a non-significant 9% reduction in falls, clearly in contrast with our results with respect to 158 patients. The differences between the studies could be explained by the fact that most of the studies tested non-slip socks as part of a multifactorial intervention, so it is difficult to determine the specific effects of non-slip socks.

One explanation for the differences in falls among groups may be justified by the “balance control” concept; balance control is based on integrating sensory information with that collected from musculoskeletal systems. Proprioception, skin sensitivity, and muscle strength are the three fundamental pillars of balance control among adults, accounting for about 60% to 70% of their total balance control, while the visual and vestibular systems contribute the rest. Despite its safety measures such as the use of Velcro or a closed part of the heel, adequate footwear, due to its volume with respect to a patient’s foot, makes it more difficult for the brain to correctly specify a foot’s dimensions as well as its exact position with respect to the environment that surrounds it; consequently, footwear can decrease sensitivity and therefore the ability to react in extreme situations. Thick soles, footwear with toes or diminished flexibility, can limit footprints. On the contrary, non-slip socks fits adequately to the contours of our feet, providing greater self-perception given their closer relationship with the surface of the entire foot and the ground, and the non-stick reinforcement bands of the sole provide better grip than footwear, helping to avoid falls caused by slipping; in addition, their smaller dimensions allow for more control [40]. It is important to add that we have used different sizes of socks to provide an adequate fit to the foot of each patient, thus helping to stop the patients from tripping over their socks.

Even though intrinsic factors have been associated with an increased risk of falls (mental status, mobility, urinary or fecal issues, and general factors including fatigue and a history of falls) [41,42,43], in our study, patients who suffered from falls did not show greater impairment.

In our study, statistical differences were observed in mean age between the non-slip socks patients and the adequate footwear group, but as the logistic regression analysis shows, age does not have a significant effect on falls. Community and long-term care studies have generally been limited to elderly patients and have found that an age above 80 years increases the risk of falling [44,45]. However, some studies point out that approximately half of inpatients fall when they are younger than 65 years old [17]. This finding may reasonably suggest the importance of other fall risk factors commonly associated with age and falling [46].

Higher mental confusion was observed in the non-slip socks group, but all falls were registered among patients who were conscious and oriented. According to different studies, there is an intricate relationship between falls, activity, and risk, suggesting that the people who are most active and those most inactive are more prone to falling. Taking into account that patients who suffered from falls were conscious and oriented in our study, physical activity could increase the risk of falling [11].

One of the most frequent reasons for admission in this group of patients was COVID-19 infection. These patients are attended by selected personnel, and sometimes, due to the use of protective clothes and equipment to prevent coronavirus infection, visits are grouped together and can be less frequent. The other more frequent reason was oncology-related symptoms; those who had the highest rates of injury might have been more prone to fall-related injuries due to anemia, thrombocytopenia, and risk of pathologic fracture [17].

An extremely important consideration in this context are falls due to iatrogenic causes: the intake of four or more drugs (antihypertensives, diuretics, benzodiazepines, and antidepressants) is considered an independent risk factor of suffering a fall. This fact is aligned with our results (more than 60% of the patients were taking more than four drugs in these groups) [47].

Slipping was the most frequent reason for falls in our study, which usually occurred whilst patients were ambulating in their rooms, and this finding is aligned with other studies [2]. One explanation for this fact could be that the patients did not use the call light because they believed that they did not need assistance. Perhaps patients need to be better educated on the effects that a new environment, decreased activity, medications, tests, and treatments can have on their energy and ability to ambulate safely.

In addition, experienced nurses evaluated the presence of preventive measures for falling, including slippery floors, and 50% of nurses reported an adequate floor surface; even though this could be a subjective appreciation, this research was performed in the same hospitalization units, so the environmental factors can be considered to be the same for all patients included in this study.

The results of this study should be interpreted with caution due to certain limitations. Some variables were not always assessed in an objective manner but rather by asking the patient about his/her health status. At times, information collected from various sources was conflicting, and so the data collector’s best judgment was used. Collecting data from different sources also introduced bias. In addition, many patients did not recall the circumstances of their fall. Even if the patients were alert and oriented, patient recall of events is often not accurate because patients will often make associations to explain the occurrence of a fall. Regarding the consequences of falls, although the reporting of falls is mandatory, it is possible that falls that do not result in injury are less likely to be reported than falls that do result in injury. Follow-up record reviews could not always be completed before the patients were discharged, resulting in missing values for several variables. Finally, our results suggest that modifiable activities and characteristics of the patients could contribute to hospital falls. Given that the falls occurred when the patients were unassisted and that ambulation prevention efforts may be helpful, encouraging patients and families to bring an assistive device from home for use in the hospital could help prevent some falls. Prevention efforts can then be targeted toward patients at high risk and address the known activities leading to falls and the characteristics contributing to patients being at higher risk.

## 5. Conclusions

The poorer relative performance of the “ideal” foot condition compared to non-slip socks in our study suggests that non-slip socks represent an adequate alternative to well-fitting rubber-soled footwear. It seems that non-slip socks could prevent falls among hospitalized patients; nevertheless, further investigations could add information.

Other variables considered to be predictors of falls, such as age, mental status, or previous risk of falling, should be analyzed in a multifactorial context.

Although most of the falls did not result in an injury, it is important to develop fall prevention programs that include assistive equipment such as non-slip socks, among other forms, and focus on modifiable activities and characteristics of a patient that could decrease hospital falls.

## Figures and Tables

**Figure 1 healthcare-11-02605-f001:**
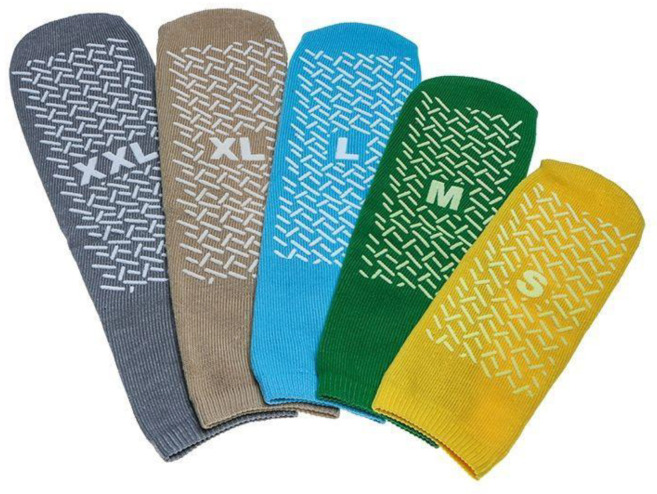
Non-slip socks.

**Figure 2 healthcare-11-02605-f002:**
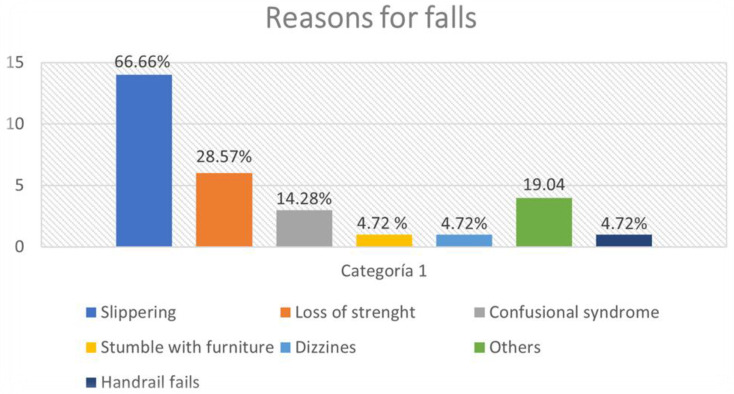
Reasons for falls.

**Figure 3 healthcare-11-02605-f003:**
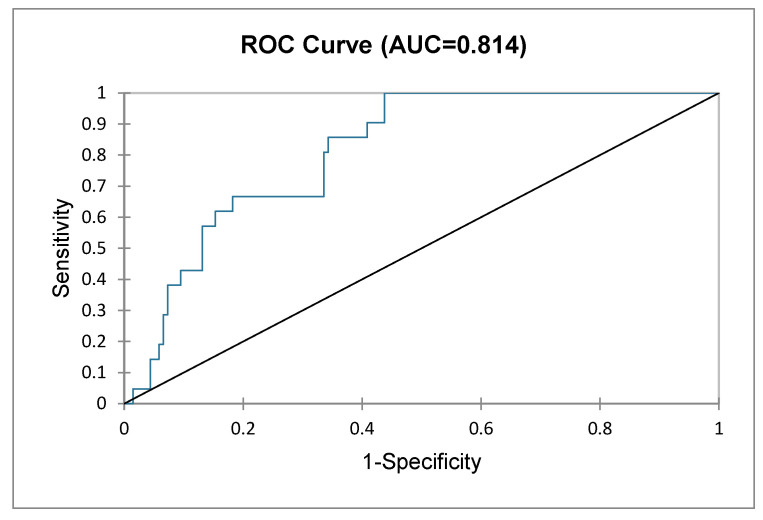
Logistic regression ROC curve.

**Table 1 healthcare-11-02605-t001:** Baseline characteristics.

	Non-Slip Socks	Adequate Footwear	*p*
Gender (N)			
Women	32	42	*p* < 0.109
Men	45	37	
Age (Mean ± STD)	76.39 ± 17.55	84.45 ± 10.75	***p* < 0.001**
Treatments (N)			
Psychotropic drugs			
NO	33	24	*p* < 0.736
YES	49	36	
Antihypertensives			
NO	28	27	*p* < 0.689
YES	49	54	
Anticoagulants			
NO	74	3	*p* < 0.341
YES	3	6	
>5 drugs			
NO	50	47	*p* < 0.373
YES	27	34	
Visual impairment (N)			
NO	57	55	*p* < 0.475
YES	20	25	
Hearing disturbances (N)			
NO	60	17	*p* < 0.327
YES	56	24	
Low extremities disabilities (N)			
NO	61	62	*p* < 0.502
Paralysis	15	19	
Paresthesia	1	0	
Mental status (N)			
Confusion	12	4	***p* < 0.027**
Conscious and oriented	65	77	
Risk of falls (N)			
High	62	66	*p* < 0.876
Mild/Moderate	12	13	
No risk	3	2	

**Table 2 healthcare-11-02605-t002:** Patient Risk factors for falls.

	*N*	Category	Frequency (N)	Frequency (%)
Psychotropic drugs	21	NO	14	66.67
		YES	7	33.33
Antihypertensives	21	NO	15	71.43
		YES	6	28.57
Anticoagulants	21	NO	16	76.19
		YES	5	23.81
>5 drugs	21	NO	8	38.01
		YES	13	61.90
Other drugs	21	NO	17	80.95
		YES	4	19.05
Visual impairment	21	NO	12	57.14
		YES	8	38.10
		UNKNOWN	1	4.76
Hearing disturbances	21	UNKNOWN	1	4.76
		NO	10	47.62
		YES	10	47.62
Lower-extremities disabilities	21	NO	10	47.62
		YES	11	52.38
Mental status	21	Confusion		
		Conscious and oriented	21	100.00

## Data Availability

Data available on request due to privacy restrictions eg privacy or ethical. The data presented in this study are available on request from the corresponding author. The data are not publicly available due to ethics and privacy laws in Spain.

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
