# Peer review of "Observational Prospective Study to Determine the Efficacy of ‘Non-Slip Socks’ vs. ‘Adequate Footwear’ Regarding the Number of Falls Observed among Admitted Patients"

_healthcare, 2023, doi:10.3390/healthcare11192605_

Round 1

Reviewer 1 Report

This work has high clinical relevance. However, some aspects should be considered before its publication.

Page 2. Line 70: The term "are" is repeated.

Page 3, Line 101: The recruited patients are over 18 years old. However, postural control, joint stability, muscle strength, and even reflexes differ between a young patient and an elderly patient, and these factors are determinants of the risk of falls. It is suggested to delimit the age range of the sample (excluding extremes) and include inclusion and exclusion criteria in this section.

Can physical condition and motor control affect the use of non-slip socks? Having better physical condition improves stability, and therefore, the use of non-slip socks would be indicated for these patients. Has this been considered when grouping the sample?

Page 3, Line 104: The authors should include images of the footwear and socks used in the study.

Page 5. Line 184: It is necessary to include the registration number of the Ethics Committee in case the reader wishes to consult it.

Page 6, Line 181: It is advisable to place the citation number next to the author, rather than at the end of the paragraph. This makes it easier to read.

Author Response

Reviewer 1

  • Page 2. Line 70: The term "are" is repeated. Corrected
  • Page 3, Line 101: The recruited patients are over 18 years old. However, postural control, joint stability, muscle strength, and even reflexes differ between a young patient and an elderly patient, and these factors are determinants of the risk of falls. It is suggested to delimit the age range of the sample (excluding extremes) and include inclusion and exclusion criteria in this section.Despite the fact  that patients over 18 could be included in our study , most of patients were over 65(134). Just 24 are under this age with a range from 58 to 64. Unfortunately is not posible at this stage to change inclusion criteria, because these were approved by Ethical Committee
  • Can physical condition and motor control affect the use of non-slip socks? Having better physical condition improves stability, and therefore, the use of non-slip socks would be indicated for these patients. Has this been considered when grouping the sample?There are not information in the literature about this fact, but we have included in the discusion the concept of propioception; non-slip socks fits adequately the contour of our foot, providing a greater self-perception given their closer relationship with the surface of the entire foot and the ground, the non-stick reinforcement bands of the sole provide a better grip than footwear helping to avoid falls caused by slipping.
  • Page 3, Line 104: The authors should include images of the footwear and socks used in the study. Added
  • Page 5. Line 184: It is necessary to include the registration number of the Ethics Committee in case the reader wishes to consult it. In Spain we don’t have an opened registry to consult this, each hospital is responsible to keep the records of the approved studies. In case that a reader needed to consult the approval of Ethics Committee, he/she can contact with either correspondence authors who can provide a copy.
  • Page 6, Line 181: It is advisable to place the citation number next to the author, rather than at the end of the paragraph. This makes it easier to read. Thank you for the suggestion we will try to do it.

Reviewer 2 Report

A lot of typos that must be corrected.

ln 125 what 95% CI?

ln 132 the software must have a version and a company that must be specified.

How was measured mental status? Define the risk of falls for every category. Discuss more the results.

You analized the patient risk factors for falls, you can perform a logistic regression so you can discuss more about them.

A lot of typos that must be corrected.

Do you present the numbers with commas or not?

Author Response

  • A lot of typos that must be corrected. Reviewed
  • ln 125 what 95% CI? To calculate the sample size, the prevalence of falls reported in Spain and the statistics in our hospital, focus on the same services during the same period in previous years, were taking into consideration; falls prevalence was estimated in 1.1 %. Assuming an alpha value of 0.05, and a margin of error of 5%, the required sample size was estimated in 150 subjects. This explanation has been added to the text
  • ln 132 the software must have a version and a company that must be specified. XLSTAT versión 2022.4.1  Addinsoft (2023). XLSTAT statistical and data analysis solution. New York, USA. https://www.xlstat.com/es. Added to the text
  • How was measured mental status? Define the risk of falls for every category. Discuss more the results. Mental status was defined following Downton scale (orientated, confused)Looking at the literature there are not evidence that mental status was a independent predictive factor of falls
  • You analized the patient risk factors for falls, you can perform a logistic regression so you can discuss more about them.
    Following your suggestion we have performed a logistic regression, the results will be added and discussed.A logistic regression was performed to model the effect of some variables on falls. We considered 2 explanatory variables for falls, one qualitative –type of footwear - and one quantitative corresponding to the age, Chi² associated with the Log ratio is lower than 0.0001, we can conclude that the variables bring a significant amount of information. In Type II analysis we can see that the variable that most influences falls is type of footwear (p = 0.000). The group of “non slip socks” has a significant negative impact (-18,102) in falls, whilts age does not have any significant effect. The classification table for the training sample shows that the observations were well classified at 86,71%.  The area under the curve for ROC curve was 0,814  showing a well-discriminating model.

Reviewer 3 Report

Thank you for submitting your paper.

There are some aspects that, in my opinion, must be clarified to consider the paper to be published

In the introduction (line 53) appears that in Spain the incidence is 1,66% but later in Madrid is 1,76‰ in 2020 when it has been worst than in previous years. It is very confusing to be understood.

One of the aspects that have been looking for is "paralysis". Paralysis means that the patient doesn´t have the possibility to walk. Why are included in this study patients without the possibility to walk? Obviously, they are at risk of falling. This must be clearly explained. What means paralysis, how have you measured it?

There is a significant difference in age. It is known that age is a risk factor for falling. It could be important to explain the relationship between age and falls in your study population.

How have you measured the risk of falls and mental status? Which scale have you used?

Which is the importance of the information provided in lines 167-169?

In line 173 you say that 52,4% of patients who have fallen "did not wear closed heel shoes". This is one of the characteristics of what you define as adequate footwear. In my opinion, if the patients are not using adequate footwear and they fall, probably it could have some relation whit your results that must be clearly explained.

In line 174 you say that patients used fastening strips in the group who have fallen. Patients with mobility restriction have fallen? In my opinion, again, these aspects are important to be discussed during the text.

In line 197 you talk about proprioception in your study and explain the loosing of proprioception produced by the shoes. But in elderly people, with such morbidities as the ones that your population has, with differences in mean age, other aspects could affect proprioception like brain injury and polyneuropathies,... You do not consider these aspects in your explanation

In your conclusions, in line 270 "predictors of injury should be considered as well as predictors of patient fall risk....". and in line 274 "encouraging patients and families to bring an assistive device from home for use in the hospital could help prevent some falls." Taking into account your study and your observations I think you must explain how can you obtain these conclusions because they look like if they are subjective suggestions, that maybe could be introduced in the discussion, but that are not conclusions that appear from your results.

It is also important to use standardized language. I don´t understand why you used mean age, average age, and medium age (?????) in the same text.

I invite you to a thorough revision of the text to adapt the discussion and conclusions to the results of your study and clarify some aspects that simplify the interpretation of it.

Author Response

  • In the introduction (line 53) appears that in Spain the incidence is 1,66% but later in Madrid is 1,76‰ in 2020 when it has been worst than in previous years. It is very confusing to be understood. It is important to take into consideration that this incidence rate of falls in hospitalized patients in Madrid is calculated per-1,000 days of stay during the period and include adult and pediatric patients from different services as medical, surgical, mental health hospi-talization, resuscitation services intensive care units (adults and pediatrics) and maternity, obstetrics and delivery services. We have included this explanation in the text
  • One of the aspects that have been looking for is "paralysis". Paralysis means that the patient doesn´t have the possibility to walk. Why are included in this study patients without the possibility to walk? Obviously, they are at risk of falling. This must be clearly explained. What means paralysis, how have you measured it? We used  Downton scale in our study to evaluated risk of falls; you are right and maybe we have to explain more in detail this scale in order to avoid confusion. In Downton scale sensory deficits are classified as none, visual impairment,hearing impairment and limb impairment, inside this group, there are several conditions which can affect, 3 or more is considered high risk of falls. Following your suggestion we have introduced changes in methods section to explain in detail Downton scale
  • There is a significant difference in age. It is known that age is a risk factor for falling. It could be important to explain the relationship between age and falls in your study population. In this sense we have performed a logistic regression to understand or predict the effect of different variables on falls. This analysis has been added to the results
  • How have you measured the risk of falls and mental status? Which scale have you used? Risk of falls was measured by Downton scale (>3)
  • Which is the importance of the information provided in lines 167-169? We used a template provided by the journal and missed some lines, there are not additional information between line 148 to 168
  • In line 173 you say that 52,4% of patients who have fallen "did not wear closed heel shoes". This is one of the characteristics of what you define as adequate footwear. In my opinion, if the patients are not using adequate footwear and they fall, probably it could have some relation whit your results that must be clearly explained. Patients don`t fit adequately their shoes. Despite the fact the shoe was considered adequate, patients use the shoes as slippers or in inappropiate way; an explanation has been added
  • In line 174 you say that patients used fastening strips in the group who have fallen. Patients with mobility restriction have fallen? In my opinion, again, these aspects are important to be discussed during the text. 2 patients had mobility restrictions both in hospital chair, one in chest and abdomen, the other one just in chest and the reason for fall was chair instability in both cases. The explanation has been added to the text
  • In line 197 you talk about proprioception in your study and explain the loosing of proprioception produced by the shoes. But in elderly people, with such morbidities as the ones that your population has, with differences in mean age, other aspects could affect proprioception like brain injury and polyneuropathies, You do not consider these aspects in your explanation. We did not collect in CRF previous conditions of the patients, so we are no table to discuss this point, but we are planning to study this relationship in our next research
  • In your conclusions, in line 270 "predictors of injury should be considered as well as predictors of patient fall risk....". and in line 274 "encouraging patients and families to bring an assistive device from home for use in the hospital could help prevent some falls." Taking into account your study and your observations I think you must explain how can you obtain these conclusions because they look like if they are subjective suggestions, that maybe could be introduced in the discussion, but that are not conclusions that appear from your results. We have reviewed the conclusions and have modified according your suggestions

It is also important to use standardized language. I don´t understand why you used mean age, average age, and medium age (?????) in the same text. Corrected

I invite you to a thorough revision of the text to adapt the discussion and conclusions to the results of your study and clarify some aspects that simplify the interpretation of it.Done

Round 2

Reviewer 1 Report

The authors have made the changes requested by the reviewer, and the article is ready for publication.

Author Response

Thank you for your response, we are happy to know that for your point of view the article is ready for publication.

Reviewer 2 Report

The article was improved.

Moderate editing of English language is required.

Author Response

Thank you for your response, we have reviewed English language of the article

Reviewer 3 Report

Thank you for your manuscript

I continue thinking that there is an incongruity in the introduction

lines 52-54 "In Spain the incidence of falls in an acute hospital, shows significant differences depending on the characteristics of each hospital. The mean incidence of falls in acute hospitals is estimated in a 1.66% with respect to the number of patients admitted" 1.66 per cent

lines 59-61 "In 2020, the incidence of falls in the hospitals of the Madrid Health Service was 59 1.76. 75% of hospitals have an incidence of falls of less than 2.38‰, with a minimum 60 result of 0.20and maximum of 4.54‰.  All are per thousand data

Are Madrid´s hospitals so much safer than the other hospitals in Spain.

Are you using two differents data for the same question.

I understood the explanation but this incongruity tends to mislead readers

All the best

Author Response

Thank you for your response. I hope we can clarify you doubts

  • Are Madrid´s hospitals so much safer than the other hospitals in Spain? No, they are not much safer.Madrid hospitals have similar incidence of falls than other hospitals in Spain. The differences in incidence of falls depends on sociodemographic characteristics of each center.

  • Are you using two differents data for the same question? Yes we are using different data sources. In the case of incidence of falls in Madrid we used official data provided by healthcare authorities which are calculated based on incidence rate of falls in hospitalized patients per 1,000 days of stay during the study period. The mean incidence of falls in acute hospitals  in Spain was extracted from a literature review based on studies in different hospitals across Spain. We don't have available official data of incidence falls in Spain because there are 17 differents healthcare systems (one in each autonomous community)